# Serum IGFBP-1 Concentration as a Predictor of Outcome after Ischemic Stroke—A Prospective Observational Study

**DOI:** 10.3390/ijms24119120

**Published:** 2023-05-23

**Authors:** Daniel Åberg, Gustaf Gadd, Katarina Jood, Petra Redfors, Tara M. Stanne, Jörgen Isgaard, Kaj Blennow, Henrik Zetterberg, Christina Jern, N. David Åberg, Johan Svensson

**Affiliations:** 1Department of Internal Medicine and Clinical Nutrition, Institute of Medicine, Sahlgrenska Academy, University of Gothenburg, Blå Stråket 5, 413 45 Gothenburg, Sweden; gustaf.gadd@vgregion.se (G.G.); jorgen.isgaard@medic.gu.se (J.I.); david.aberg@medic.gu.se (N.D.Å.); johan.svensson@medic.gu.se (J.S.); 2Region Västra Götaland, Department of Specialist Medicine, Sahlgrenska University Hospital, Blå Stråket 5, 413 45 Gothenburg, Sweden; 3Region Västra Götaland, Department of Acute Medicine and Geriatrics, Sahlgrenska University Hospital, Blå Stråket 5, 413 45 Gothenburg, Sweden; 4Department of Clinical Neuroscience, Institute of Neuroscience and Physiology, Sahlgrenska Academy, University of Gothenburg, 405 30 Gothenburg, Sweden; katarina.jood@neuro.gu.se (K.J.); petra.redfors@vgregion.se (P.R.); christina.jern@neuro.gu.se (C.J.); 5Region Västra Götaland, Department of Neurology, Sahlgrenska University Hospital, Blå Stråket 5, 413 45 Gothenburg, Sweden; 6Department of Laboratory Medicine, Institute of Biomedicine, Sahlgrenska Academy, University of Gothenburg, 405 30 Gothenburg, Sweden; tara.stanne@gu.se; 7Region Västra Götaland, Clinical Neurochemistry Laboratory, Sahlgrenska University Hospital, 431 80 Mölndal, Sweden; kaj.blennow@neuro.gu.se (K.B.); henrik.zetterberg@clinchem.gu.se (H.Z.); 8Department of Psychiatry and Neurochemistry, Institute of Neuroscience and Physiology, Sahlgrenska Academy, University of Gothenburg, 431 80 Mölndal, Sweden; 9Department of Neurodegenerative Disease, UCL Institute of Neurology, Queen Square, London WC1N 3BG, UK; 10UK Dementia Research Institute, University College London (UCL), London WC1E 6BT, UK; 11Hong Kong Center for Neurodegenerative Diseases, Clear Water Bay, Hong Kong, China; 12Wisconsin Alzheimer’s Disease Research Center, University of Wisconsin School of Medicine and Public Health, University of Wisconsin-Madison, Madison, WI 53706-1380, USA; 13Region Västra Götaland, Department of Genetics and Genomics, Sahlgrenska University Hospital, Blå Stråket 5, 413 45 Gothenburg, Sweden; 14Region Västra Götaland, Department of Internal Medicine, Skaraborg Central Hospital, 541 42 Skövde, Sweden

**Keywords:** all-cause mortality, body mass index (BMI), insulin-like growth factor-binding protein-1 (IGFBP-1), ischemic stroke, modified Rankin Scale (mRS), mortality, National Institutes of Health Stroke Scale (NIHSS)

## Abstract

Insulin-like growth factor-binding protein-1 (IGFBP-1) regulates insulin-like growth factor-I (IGF-I) bioactivity, and is a central player in normal growth, metabolism, and stroke recovery. However, the role of serum IGFBP-1 (s-IGFBP-1) after ischemic stroke is unclear. We determined whether s-IGFBP-1 is predictive of poststroke outcome. The study population comprised patients (n = 470) and controls (n = 471) from the Sahlgrenska Academy Study on Ischemic Stroke (SAHLSIS). Functional outcome was evaluated after 3 months, 2, and 7 years using the modified Rankin Scale (mRS). Survival was followed for a minimum of 7 years or until death. S-IGFBP-1 was increased after 3 months (*p* < 0.01), but not in the acute phase after stroke, compared with the controls. Higher acute s-IGFBP-1 was associated with poor functional outcome (mRS score > 2) after 7 years [fully adjusted odds ratio (OR) per log increase 2.9, 95% confidence interval (CI): 1.4-5.9]. Moreover, higher s-IGFBP-1 after 3 months was associated with a risk of poor functional outcome after 2 and 7 years (fully adjusted: OR 3.4, 95% CI: 1.4–8.5 and OR 5.7, 95% CI: 2.5–12.8, respectively) and with increased mortality risk (fully adjusted: HR 2.0, 95% CI: 1.1–3.7). Thus, high acute s-IGFBP-1 was only associated with poor functional outcome after 7 years, whereas s-IGFBP-1 after 3 months was an independent predictor of poor long-term functional outcome and poststroke mortality.

## 1. Introduction

Insulin and insulin-like peptides are essential for glucose metabolism throughout the life span and regulate the growth, metabolism, and regeneration of both the peripheral tissues and the central nervous system (CNS) [1,2,3,4]. Furthermore, insulin and insulin-like growth factor-I (IGF-I) are altered in ischemic stroke (henceforth stroke), and these alterations are associated with recovery after stroke [5,6,7,8,9,10]. Insulin-like growth factor-binding protein-1 (IGFBP-1) is one of six high-affinity IGFBPs which modulate the bioavailability of IGF-I [2,3]. IGFBP-1 is mainly synthesized in the liver and is induced by glucocorticoids, hypoxia, stress, and starvation [3]. The role of IGFBP-1 in these conditions is to bind IGFs, thereby inhibiting IGF-I activity [3]. In the opposite direction, hepatic IGFBP-1 is reduced by insulin from the portal circulation, resulting in increased IGF-I activity during periods of high insulin production [11].

Although there are conflicting results, multiple clinical studies suggest that IGFBP-1 may be involved in metabolism and diseases related to diabetes mellitus (henceforth: diabetes) [4,12]. For example, in patients with type 2 diabetes (n = 74), low serum IGFBP-1 (s-IGFBP-1) was correlated with cardiovascular risk factors [13]. Another study demonstrated that IGFBP-1 increased β-cell regeneration experimentally [14]. In this extensive study, it was also reported that high baseline s-IGFBP-1 levels in a cohort without baseline diabetes (n = 1190) were associated with a reduced risk of developing type 2 diabetes over 10 years [14]. However, in a prospective cohort study of older men (n = 3983), high s-IGFBP-1 was associated with an increased risk of both all-cause mortality and cardiovascular mortality [15]. Finally, in older women (n = 338), both high and low s-IGFBP-1 levels, as compared with normal s-IGFBP-1, were associated with an increased risk of all-cause mortality [16].

In coronary heart disease, several studies have evaluated the usefulness of IGFBP-1 as a prognostic marker. In a small study in the acute phase after myocardial infarction (MI), s-IGFBP-1 was significantly lower in MI patients (n = 34) than in healthy controls (n = 17) [17], indicating that low s-IGFBP-1 may be harmful. In contrast, in studies examining the prognostic value of IGFBP-1 for long-term outcome after MI, high s-IGFBP-1 at admission was associated with future hospitalization for heart failure [18], long-term mortality [19], and both cardiovascular morbidity and long-term mortality [20].

There are few reports on IGFBP-1 and the risk of stroke. The results of two genetic studies have suggested that a single-nucleotide polymorphism (rs1874479) in the *IGFBP1* gene is associated with increased stroke risk [21,22]. Furthermore, in a subcohort of the Cardiovascular Health Study (n = 2026), baseline fasting plasma IGFBP-1 was similar in adults aged 65 years or older that later developed stroke compared with control subjects, and IGFBP-1 did not predict the risk of stroke [23]. In another subcohort of the Cardiovascular Health Study (n = 2268), a higher two-hour s-IGFBP-1 post-oral glucose tolerance test, but not fasting s-IGFBP-1, was associated with increased risk of major health events (which included stroke) and mortality [24]. However, to our knowledge, the relationship between s-IGFBP-1 and poststroke functional outcome and survival has previously not been examined in a stroke population.

In summary, little is known whether s-IGFBP-1 can predict the outcome after stroke. Therefore, we used the Sahlgrenska Academy Study on Ischemic Stroke (SAHLSIS) cohort to investigate whether s-IGFBP-1 levels in the acute phase or 3 months after stroke are associated with outcome after stroke in terms of functional independence and mortality. We hypothesized that high s-IGFBP-1 would predict worse stroke outcome as previously seen after MI.

## 2. Results

### 2.1. Baseline Characteristics and s-IGFBP-1 in Patients and Controls

Characteristics of the study participants are given in Table 1. As reported previously [7,25], patients had a higher frequency of hypertension, atrial fibrillation, diabetes mellitus, and smoking, as well as higher high-sensitivity C-reactive protein (hsCRP) compared with the controls. In contrast, low-density lipoprotein (LDL) levels and body mass index (BMI) did not differ significantly between groups (Table 1). S-IGFBP-1 was similar to controls in the acute phase, whereas after 3 months, it was higher in the stroke patients compared with the controls (*p* < 0.01, Table 1). In patients, s-IGFBP-1 was lower in the acute phase than after 3 months (*p* < 0.001). Moreover, we analyzed whether there was a correlation between the day after stroke onset when the blood sample was collected and the median level of s-IGFBP-1, and there was no such correlation in the acute phase (Figure 1). However, individual multiple time points of the daily s-IGFBP-1 levels in the acute phase were not available.

### 2.2. S-IGFBP-1 in Stroke Etiology Subtypes

S-IGFBP-1 levels in the etiological subtypes are given in Table 1. S-IGFBP-1 levels in the acute phase after stroke were similar in all stroke subtypes compared with the controls. In contrast, 3-month s-IGFBP-1 was increased in small artery occlusion and cardioembolic stroke (*p* < 0.01 and *p* < 0.05, respectively, vs. the control group) (Table 1).

### 2.3. Bivariate Correlations

S-IGFBP-1 was correlated with some risk factors as exhibited in Table 2. S-IGFBP-1 was inversely correlated with Homeostatic model assessment of IR (HOMA-IR), i.e., insulin resistance (r = −0.33, *p* < 0.001 acutely and r = −0.26, *p* < 0.001 after 3 months), and BMI (r = −0.25, *p* < 0.001 acutely and r = −0.28, *p* < 0.001 after 3 months).

Additionally, some other statistically significant correlations were observed with small or very small magnitudes, i.e., r < 0.2 (Table 2), and are listed as follows: acute s-IGFBP-1, but not 3-month s-IGFBP-1, correlated inversely with National Institutes of Health Stroke Scale (NIHSS), i.e., stroke severity. Additionally, acute but not 3-month s-IGFBP-1 correlated inversely with LDL. S-IGFBP-1 showed a small correlation with older age in patients in the acute phase, after 3 months, and in controls. Additionally, s-IGFBP-1 was higher in patients with diabetes than in patients without diabetes in the acute phase, as well as after three months. Furthermore, s-IGFBP-1 acutely and after 3 months correlated with current smoking. S-IGFBP-1 was not correlated with hsCRP or hypertension.

### 2.4. S-IGFBP-1 and Functional Outcome

S-IGFBP-1 levels in the acute phase were not associated with the risk of poor functional outcome evaluated through modified Rankin Scale (mRS), i.e., mRS 3–6 after 3 months (not shown) or 2 years (Table 3). In contrast, acute s-IGFBP-1 was associated with the risk of poor functional outcome after 7 years [crude odds ratio (OR) 2.17, 95% confidence interval (CI): 1.24–3.39; fully adjusted OR 2.88, 95% CI: 1.40–5.92]. However, this association was attenuated by the exclusion of patients that died (mRS 6, n = 60), with a crude OR 1.67, 95% CI: 0.82–3.41; while in the fully adjusted model there was still a significant association (OR 3.57, 95% CI: 1.36–9.40) (lower panels, Table 3).

S-IGFBP-1 after 3 months was associated with poor functional outcome after 2 years (crude OR 3.11, 95% CI: 1.57–6.17; fully adjusted OR 3.41, 95% CI: 1.37–8.52, Table 3) and after 7 years (crude OR 5.22, 95% CI: 2.69–10.1; fully adjusted OR 5.69, 95% CI: 2.53–12.8, Table 3). Additionally, after excluding mRS 6, the latter association remained significant (crude OR 4.24, 95% CI: 1.91–9.39; fully adjusted OR 5.12, 95% CI: 1.82–14.4) (lower panels, Table 3).

### 2.5. Poststroke Long-Term Mortality and s-IGFBP-1

The median levels of s-IGFBP-1 were 4.80 ng/mL in the acute phase and 6.13 ng/mL after 3 months. Cumulative survival curves showed that acute s-IGFBP-1 only had a non-significant tendency to associate with mortality (above median s-IGFBP-1 vs. below median s-IGFBP-1, log-rank test: *p* = 0.055, not shown). In contrast, high s-IGFBP-1 after 3 months was associated with an increased mortality risk (above median s-IGFBP-1 vs. below median s-IGFBP-1, log-rank test: *p* = 0.007; Figure 2).

In further analyses, Cox proportional hazards regression showed that high acute s-IGFBP-1 was associated with increased risk of mortality [crude hazard ratio (HR) 2.09, 95% CI 1.22–3.56], but this association lost significance after adjustment for covariates (fully adjusted: HR 1.38, 95% CI: 0.78–2.47) (Table 4). However, s-IGFBP-1 after 3 months was associated with increased mortality risk after full adjustment for covariates (crude HR 3.26, 95% CI: 1.82–5.84; fully adjusted: HR 2.00, 95% CI: 1.07–3.73) (Table 4).

### 2.6. Poststroke Long-Term Mortality and s-IGFBP-1 in Stroke Subtypes

In alignment with the other Cox proportional hazards regression analyses, we performed analyses of whether log IGFBP-1 was associated with mortality in the main subtypes of ischemic stroke. Due to the small number of participants and events in each subgroup, we only adjusted for two covariates [26]. Acute s-IGFBP-1 was associated with mortality in large vessel disease (crude HR 5.77, 95% CI 2.00–16.6 and adjusted HR 4.86, 1.61–14.7, respectively, Table 5A).

Regarding IGFBP-1 after 3 months, it was associated with mortality in cardioembolic stroke (crude HR 3.80, 95% CI 1.32–10.9, and adjusted HR 4.76, 95% CI 1.42–16.0, Table 5B) and cryptogenic stroke (crude HR 5.29, 95% CI 1.27–22.0, and adjusted HR 5.53, 95% CI 1.12–27.3, Table 5B).

## 3. Discussion

To our knowledge, s-IGFBP-1 levels and their association with stroke outcome have not been investigated before in a stroke population. The present study found that s-IGFBP-1 levels were higher in stroke patients after 3 months, but not in the acute phase after stroke, compared with controls. Higher acute s-IGFBP-1 was associated with poor functional outcome (mRS > 2) after 7 years, and higher s-IGFBP-1 after 3 months was associated with increased risk of poor functional outcome after 2 and 7 years. Moreover, 3-month s-IGFBP-1, but not acute s-IGFBP-1, was associated with an increased risk of poststroke mortality.

In the present study, s-IGFBP-1 in the acute phase of stroke was similar to that in the controls. Although longitudinal measurements of s-IGFBP-1 have previously not been performed in stroke patients, this finding is in some accordance with the earlier observation of unchanged baseline fasting plasma IGFBP-1 in older adults that later developed stroke compared with controls [23]. In a small study of 34 patients and 17 controls in the acute phase after MI, levels of IGFBP-1 were significantly lower in the MI patients than in the healthy controls [17]. In line with this, our patients had a non-significant tendency of lower s-IGFBP-1 values acutely after stroke, albeit after 3 months, s-IGFBP-1 was prominently higher than in the controls.

A significant finding of the present study is that high 3-month s-IGFBP-1 was independently associated with an increased risk of poor functional outcome after 2 and 7 years. This association was not driven by mortality (Table 3), although an increased risk of long-term mortality was also significantly associated with high 3-month s-IGFBP-1 (Table 4). In terms of mortality, we observed an association between 3-month s-IGFBP-1 and all-cause mortality, i.e., all mortality including not only mortality due to stroke or other cardiovascular mortality. These relations with mortality are in line with the previous associations between high s-IGFBP-1 and particularly increased risk of all-cause mortality in elderly individuals [15,16] as well as the earlier seen associations with long-term outcomes, particularly long-term all-cause mortality, after MI [18,19,20]. Thus, in line with our results, the data from other cohorts not suffering from stroke indicate that high s-IGFBP-1 is associated with an increased risk of all-cause mortality and not only cardiovascular mortality [15,16,18,19]. Interestingly, IGFBP-1 has also been related to diabetes-related diseases, catabolism, and cancer [4], and further studies are needed to explore the specific types of mortality associated with s-IGFBP-1 levels. In summary, our results suggest that s-IGFBP-1 after 3 months, which possibly resembles s-IGFBP-1 under poststroke steady-state conditions, is an independent predictor of the long-term outcome after stroke regarding functional independence and all-cause mortality.

In contrast to the significant associations between 3-month s-IGFBP-1 and long-term outcome poststroke, high acute s-IGFBP-1 was not associated with functional outcome after 3 months and 2 years, and the association between high acute s-IGFBP-1 and increased risk of mortality lost statistical significance after full adjustment for covariates. However, there was a significant association between high acute s-IGFBP-1 and increased risk of poor functional outcome (mRS 3–6) after 7 years. This association remained significant after full adjustment for covariates, although it was attenuated by the exclusion of patients that died (mRS 6). It can be noted that the directions of the non-significant associations between acute s-IGFBP-1 and functional outcome or mortality were similar to the significant 3-month s-IGFBP-1 associations. However, overall, acute s-IGFBP-1 was a weaker predictor of functional outcome following stroke.

S-IGFBP-1 and outcome were measured and followed up at several time points in this study, and this longitudinal aspect enabled us to analyze which poststroke phase s-IGFBP-1 was more critical. The different stages and nominal division of the timing aspect of stroke recovery have been debated, but a consensus has been established [27]. In the context of our findings, s-IGFBP-1 in the acute phase was not related to functional outcome after 3 months or 2 years. However, high acute s-IGFBP-1 was significantly related to poor functional outcome 7 years poststroke, even when patients who died were excluded, which may implicate that s-IGFBP-1 was of a lower significance in the acute to early subacute phase of stroke. Compared with the more robust associations between 3-month s-IGFBP-1 and long-term poststroke outcomes, this suggests that s-IGFBP-1 measured in the latter stages of stroke recovery is of more value as a prognostic marker of poststroke outcome.

In addition to in elderly populations [15,16] and MI cohorts [18,19,20], IGFBP-1 has been examined in various populations or outcomes that are either difficult to compare or sometimes with conflicting results. Furthermore, IGFBP-1 has been shown to play an important role in metabolism and diabetes-related diseases [4,12]. In people with diabetes, low levels of IGFBP-1 predicted future cardiovascular risk [13], and in a prospective study of non-diabetics at baseline, high levels of IGFBP-1 reduced the risk of developing insulin resistance or type 2 diabetes during a 10-year time span [14]. However, both these studies [13,14] lacked data regarding functional outcome and mortality. In the present study, there were numerous, although mostly relatively small, correlations between s-IGFBP-1 and metabolic risk factors, including diabetes, HOMA-IR, and BMI. However, this contrasted with the absence of a correlation between s-IGFBP-1 and hsCRP. This may indicate that s-IGFBP-1 reflects metabolic changes that influence long-term outcome and not an effect driven by an inflammatory response. This also implicates s-IGFBP-1 as a biomarker with less of a role in acute cell death, inflammation, and scarring and more of a role in the recovery in the chronic stages poststroke, and that s-IGFBP-1 may have a predictive role, especially in long-term survival. In our models, we adjusted for diabetes, BMI, and LDL without any major attenuations in the effect size. However, in the context of IGFBP-1 and all-cause mortality, IGFBP-1 has been proposed to influence mortality through cancer [4,28], and we cannot rule out that poststroke cancer may confound the effect on mortality.

The temporal profile of the increase in 3-month s-IGFBP-1, which is quite late in the subacute poststroke phase rather than acute, could also be secondary to other types of yet unknown mechanisms. Interestingly, it resembles the results of an earlier substudy of SAHLSIS, which evaluated neurofilament light chain (NfL), a marker of neuroaxonal damage [29]. The highest s-NfL was observed in the subacute phase, and high s-NfL was associated with poor outcomes, with the strongest association between 3-month s-NfL and poststroke outcome [29].

We found different levels of s-IGFBP-1 in the stroke subtypes after 3 months, as s-IGFBP-1 was increased in small vessel occlusion and cardioembolic stroke. Additionally, higher s-IGFBP-1 was associated with increased risk of mortality in some subtypes of stroke. The temporal profile of s-IGFBP-1, i.e., whether the sample was taken acutely or after 3 months, seemed to alter the subtype of stroke in which s-IGFBP-1 was associated with long-term mortality. Different mechanisms of action may have different impacts on subtypes of ischemic stroke [30]. Therefore, the predictive value of s-IGFBP-1 may be higher in some stroke subtypes, but this remains to be elucidated in further studies.

### Strengths and Limitations

Our study had a relatively large study sample, including consecutive and well-characterized ischemic stroke patients, as well as population-based controls. Additionally, the high hospitalization rate, nearly 90% for stroke patients < 70 years in Sweden [31], likely reduced selection bias. In terms of the multivariate regression models for functional outcome and mortality, statistical power was retained with reasonable strength [26] when adjusting for the limited covariates that we included. Other advantages of our study are that the blood samples were drawn both in the acute phase and 3 months after stroke, and the blood samples from the two time points were analyzed simultaneously. However, our study also has significant limitations. First, our study did not include a replication cohort. Second, the acute blood samples were not drawn immediately after stroke onset, i.e., they were taken after a median of 4 days, and may therefore reflect the stress response in the acute phase of stroke. Third, we cannot exclude the possibility that associations during the follow-up were influenced by unaccounted health- and treatment-related factors. Our analyses did not include prestroke data, other than from some self-reported variables such as previous smoking, thus diminishing the primary prevention perspective and possibilities for prestroke confounder analysis. We included relatively young (<70 years old) Caucasian stroke patients and controls, which limits the generalizability of the results from a global perspective. Although our study has an advantage of long-term data regarding functional outcome and mortality, the study was initiated in an era with different treatment of ischemic stroke, i.e., today, thrombolysis is more common, antiplatelet and anticoagulant regimens have been altered, and thrombectomy is now routinely used. These secular trends of treatment of stroke modulates long-term outcome as well as mortality after stroke and may limit the conclusions from a contemporary and future perspective. Finally, s-IGFBP-1 levels in stroke subtypes must be interpreted cautiously, as the number of patients in these subgroups was limited (n = 55–136), and in particular, the analyses of mortality in Table 5 need to be interpreted with considerable caution since multivariate analyses are restricted in small subgroups [26].

## 4. Materials and Methods

### 4.1. Patients and Controls

The design of SAHLSIS has previously been reported [7,25]. Briefly, patients with first-ever or recurrent acute ischemic stroke before the age of 70 years were recruited. All patients included in the present study were enrolled consecutively at four Stroke Units in western Sweden between 1998 and 2003. The controls were randomly selected from a population-based health survey or the Swedish Population Register to match the patients regarding age (<1 year), sex, and area of residence. In total, there were 600 patients and 600 controls. After the exclusion of all participants without adequate blood samples for the determination of IGFBP-1, 470 patients and 471 controls were available for further analysis.

### 4.2. Classification of Stroke Etiology

Stroke etiology was classified using the Trial of Org 10172 in Acute Stroke Treatment (TOAST) criteria [32] into the subtypes of large artery atherosclerosis (LAA), small artery occlusion (SAO), cardioembolic (CE) stroke, cryptogenic stroke (when no cause was identified despite extensive evaluation), other determined cause of stroke, and undetermined stroke (incomplete evaluation or more than one etiology identified).

### 4.3. Stroke Severity and Functional Outcome

In stroke patients, initial stroke severity was initially assessed by the Scandinavian Stroke Scale (SSS). The SSS is similar to the now more commonly used National Institutes of Health Stroke Scale (NIHSS) [33], with the most crucial difference being that the scales are inverse. We recalculated the SSS scores to NIHSS scores using a validated algorithm: NIHSS = 25.68 − 0.43 × SSS [34]. Functional outcome was evaluated by the modified Rankin Scale (mRS), graded 0-6, where 0 is no disability, 5 is a severe disability, and 6 is death. Evaluation of mRS was performed after 3 months and after 2 and 7 years. If not otherwise specified, the mRS score was dichotomized for death or dependency, i.e., poor functional outcome (mRS 3–6) versus favorable functional outcome (mRS 0–2).

### 4.4. Assessment of Covariates and Ethical Considerations

Body mass index (BMI, calculated as kg/m^2^) and data on hypertension, diabetes mellitus, smoking, and atrial fibrillation at inclusion were recorded as described previously [25,35]. Diabetes was defined as receiving diet treatment or medication for diabetes or alternatively, fasting plasma glucose ≥ 7.0 mmol/L or fasting blood glucose ≥ 6.1 mmol/L for at least two occasions [36]. Among patients, measurements performed at the 3-month follow-up were used for the definition of diabetes and hypertension. Hypertension was defined by pharmacological treatment for hypertension, systolic blood pressure ≥ 160 mmHg, or diastolic blood pressure ≥ 90 mmHg [25]. Smoking history was recorded as current smoking versus never smoking or smoking cessation at least one year before inclusion in the study.

This study was conducted under the 1964 Helsinki Declaration and subsequent amendments or comparable ethical standards. All participants or their next of kin provided written informed consent. The study was approved by the Ethics Committee of the University of Gothenburg.

### 4.5. Biochemical Analysis

Venous blood samples were collected in the acute phase (median 4 days, range 1–10 days after the index stroke) and at the 3-month follow-up (median 101, range 85–125 days) in stroke patients and once in controls. Thus, in the patients, s-IGFBP-1 was assessed both in the acute phase of ischemic stroke and in the subacute phase after 3 months, the latter possibly resembling s-IGFBP-1 under poststroke steady-state conditions. Blood sampling was performed between 08:30 and 10:30 a.m. after an overnight fast of > 8 h. Serum was isolated within 2 h by centrifugation at 2000× *g* at 4 °C for 20 min and stored at −80 °C before assay. S-IGFBP-1 was measured by an enzyme-linked immunosorbent assay (ELISA) using a commercial kit from Mediagnost, Reutlingen, Germany. The measurements of s-IGFBP-1 were performed at the same time point. The inter-assay coefficient of variation (CV) was 14.6%, and the intra-assay CV was 4.5%. Due to the relatively high inter-assay variation for s-IGFBP-1, a correction factor was used, obtained by having three original serum samples in each 96-well plate. All blood and plasma concentrations of glucose and low-density lipoprotein cholesterol (LDL) were analyzed using standardized methods at the Department of Clinical Chemistry at the Sahlgrenska University Hospital. High-sensitivity CRP (hsCRP) was analyzed in serum by a solid-phase chemiluminescent immunometric assay on Immulite 2000 (Diagnostic Products Corp., Los Angeles, CA, USA) with the manufacturer’s reagents as directed.

### 4.6. Statistical Evaluation

We used SPSS version 28.0 (IBM Corp., Armonk, NY, USA). The distribution of s-IGFBP-1 was skewed according to the Kolmogorov–Smirnov test. Therefore, between-group differences were analyzed using ANOVA regarding continuous variables in Table 1 and chi-square tests for categorical variables in Table 1. We used the Kruskal–Wallis test to test between-group differences in Figure 1. Correlation coefficients were calculated using Spearman correlation analysis. Furthermore, in the multivariate analyses (binary logistic regression or Cox proportional hazards regression), we logarithmically transformed s-IGFBP-1. According to the Kolmogorov–Smirnov test, s-IGFBP-1 was normally distributed after it had been logarithmically transformed.

To investigate whether s-IGFBP-1 was associated with poor functional outcome (mRS 3–6), we performed binary logistic regression analysis. As a sensitivity analysis, we also carried out an additional analysis without mRS 6, i.e., without the dead, to exhibit whether the potential poor functional outcome was driven by functional independence or might be explained by an increased risk of death. We calculated odds ratios (ORs) and 95% confidence intervals (CIs) per log increase in s-IGFBP-1 (one log unit representing a tenfold increase). Furthermore, we evaluated whether s-IGFBP-1 (per log increase) was associated with mortality by calculating hazard ratios (HRs) and 95% CIs using Cox proportional hazards regression analysis. We evaluated all-cause mortality; the follow-up time was minimum 7 years or until death. The mean follow-up for death was 11.6 years (95% CI:11.3–11.9 years). Additionally, Kaplan–Meier survival curves were constructed to exhibit survival rates in different levels of IGFBP-1, i.e., above the median of s-IGFBP-1 compared to below the median of s-IGFBP-1, which is exhibited in Figure 2.

To examine the independent effect of s-IGFBP-1 on functional outcome (Table 3) and mortality (Table 4), adjustments were made for covariates related to stroke severity and well-known metabolic and cardiovascular risk factors. Hence, in Model A, we adjusted for age and sex. Model B: age, sex, cardiovascular risk factors (BMI, hypertension, LDL, smoking, and diabetes). Model C: age, sex, cardiovascular risk factors, and stroke severity. Additionally, as LDL had the most missing values (81/470 missing in the acute phase and 53/470 missing after 3 months), the mean LDL was imputed to replace the missing LDL values in the regression analyses. As hsCRP was not correlated with s-IGFBP-1 and did not alter the multivariate models, it was not included in the regression analyses. To examine the independent effect of s-IGFBP-1 on mortality in the four main subtypes of the TOAST classification (Table 5), where the number of the events in each group was considerably lower, adjustments had to be restricted to two covariates [26]. Hence, in Table 5, we only adjusted for age and sex. *P*-values are stated if <0.10; *p* < 0.05 was considered statistically significant.

## 5. Conclusions

This study investigated the association between s-IGFBP-1 levels and poststroke outcome in a stroke population, which has not been performed previously. S-IGFBP-1 levels were higher in ischemic stroke patients after 3 months, but not in the acute phase after stroke, compared with controls. The observed associations suggest that 3-month s-IGFBP-1 rather than acute s-IGFBP-1 is a prognostic marker of long-term functional outcome and poststroke mortality. Future studies should focus on investigating s-IGFBP-1 in relation to temporal profiles with repeated blood samples located in the subacute phase after ischemic stroke, and regarding modalities of outcome in the long-term perspective and preferably also include data on the cause of death.

## Figures and Tables

**Figure 1 ijms-24-09120-f001:**
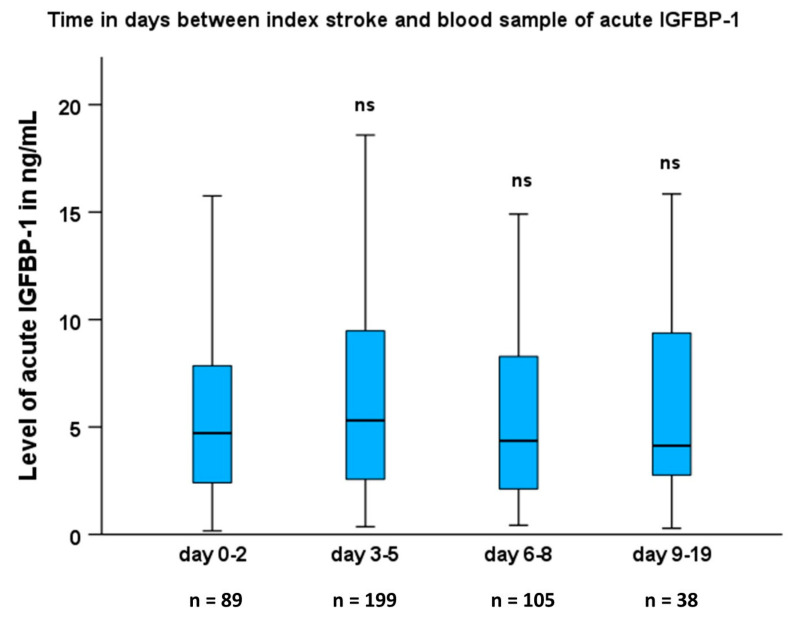
Levels of acute serum IGFBP-1 (s-IGFBP-1) in relation to which poststroke day it was sampled. Boxplots of s-IGFBP-1 in the acute phase at different time points after stroke onset. Values in the box plots are provided as medians (horizontal lines), 25–75th percentiles (boxes), and ranges (whiskers). Comparisons using the non-parametric Kruskal–Wallis test showed no difference between days 0–2 vs. days 3–5, days 6–8, and days 9–19. Thus, there were no significant changes in the levels of s-IGFBP-1 related to when the blood sample was taken. ns, not significant.

**Figure 2 ijms-24-09120-f002:**
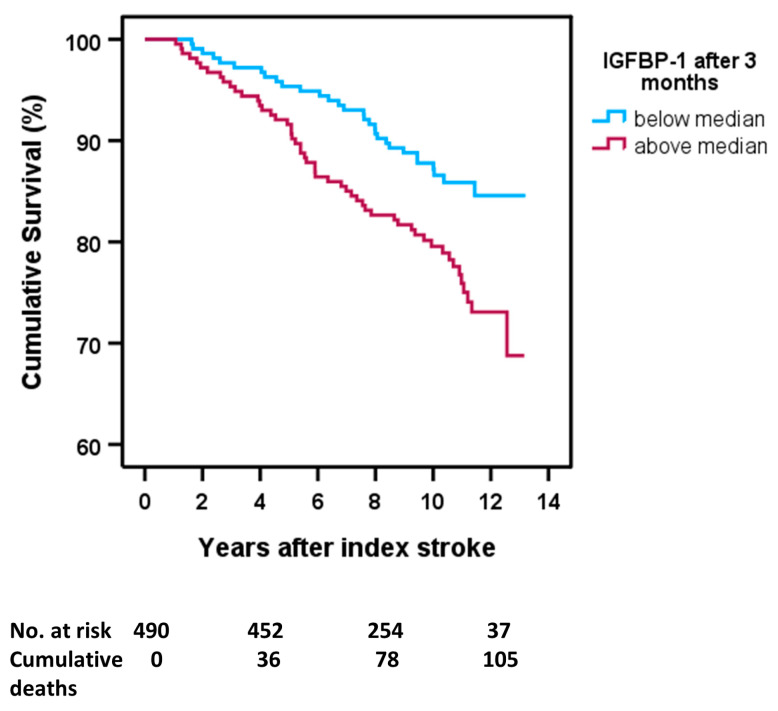
High serum IGFBP-1 (s-IGFBP-1) in samples collected 3 months after ischemic stroke is associated with an increased mortality risk. Kaplan–Meier survival curves for the risk of all-cause mortality are shown for higher versus lower than the median of 3 months s-IGFBP-1. Lower than the median s-IGFBP-1 is displayed as a blue line, and above the median of s-IGFFBP-1 is displayed as a purple line. Numbers at risk and cumulative mortality are presented at the index stroke and after 4-, 8-, and 12 years poststroke. Log-rank test for above vs. below median of IGFBP-1 3 months after IS: *p* = 0.007.

**Table 1 ijms-24-09120-t001:** Baseline characteristics in ischemic stroke patients and healthy controls as well as in stroke subtypes.

	Controls (n = 471)	Missing Values (n)	Patients (n = 470)	Missing Values (n)	LAA (n = 55)	SAO (n = 96)	Cardioembolic Stroke (n = 77)	Cryptogenic Stroke (n = 136)
Age (years)	56.8 ± 0.46	(0)	56.8 ± 0.46	(0)	58.3 ± 1.00	59.2 ± 0.70 *	58.4 ± 1.22	54.5 ± 0.91 *
Male sex, %	64	(0)	64	(0)	70	64	66	60
Hypertension, %	40	(1)	62 ***	(6)	58 *	73 ***	54 *	40 ***
Diabetes mellitus, %	6	(2)	20 ***	(0)	36 ***	26 ***	19 ***	13 **
Current smoker, %	18	(0)	40 ***	(3)	56 ***	45 ***	39 ***	39 ***
hsCRP, mg/L	3.01 ± 0.26 ^a^	(1)	10.7± 1.04 ***	(19)	12.5 ± 2.66 ***	4.46 ± 0.48 *	17.6 ± 3.46 ***	7.80 ± 1.55 ***
LDL, ng/nL	3.32 ± 0.04 ^a^	(3)	3.34 ± 0.05	(81)	3.56 ± 0.16	3.50 ± 0.11	3.01 ± 0.13 **	3.32 ± 0.08
HOMA-IR	2.06 ± 0.12 ^a^	(5)	4.84 ± 0.26 ***	(42)	6.20 ± 0.90 ***	5.13 ± 0.77 ***	4.58 ± 0.57 ***	3.70 ± 0.25 ***
BMI, kg/m^2^	26.6 ± 0.19 ^a^	(1)	26.6 ± 0.20	(12)	26.9 ± 0.66	27.1 ± 0.45	26.5 ± 0.52	26.4 ± 0.33
Atrial fibrillation, %	1	(38)	11 ***	(52)	2	2	54 ***	2
Oral anticoagulation (OAC), %	NA	NA	11	(3)	13	2	29	2
NIHSS score baseline	NA	NA	5.41 ± 0.27	(1)	6.70 ± 0.86	3.23 ± 0.27	6.82 ± 0.83	5.25 ± 0.47
mRS score 3 months	NA	NA	1.82 ± 0.06	(29)	2.20 ± 0.16	1.31 ± 0.10	2.11 ± 0.16	1.75 ± 0.09
mRS score 2 years	NA	NA	1.89 ± 0.06	(4)	2.43 ± 0.25	1.42 ± 0.10	2.22 ± 0.18	1.70 ± 0.10
mRS score 7 years	NA	NA	2.52 ± 0.10	(95)	3.46 ± 0.35	1.84 ± 0.20	3.30 ± 0.17	2.14 ± 0.17
s-IGFBP-1, acute (ng/mL), geometric mean	5.04 ± 0.31 ^a^	(0)	4.66 ± 0.37	(0)	4.43 ± 1.54	5.82 ± 0.64	5.32 ± 1.00	4.38 ± 0.71
s-IGFBP-1, 3 months (ng/mL), geometric mean	NA	NA	6.07 ± 0.48 **	(40)	5.15 ± 1.31	7.02 ± 1.02 **	7.16 ± 1.38 *	5.59 ± 0.96

Data are shown as means (±SEM), for insulin-like growth factor-binding protein-1 (IGFBP-1) values as geometric means (±SEM), or for categorical variables as %. Values are given for controls, the total stroke cohort, and the four major etiological stroke subtypes according to TOAST. The remaining patients who had either other determined causes (n = 37) or undetermined causes (n = 83) are not shown above. Differences compared to the control group were examined using ANOVA for continuous variables and the χ^2^-test for categorical variables (sex, hypertension, diabetes, smoking, atrial fibrillation, and oral anticoagulation). Furthermore, for each of the stroke etiologies, the values are compared to the controls. Numbers (n) of cases and controls as well as the main stroke subtypes are given in the headings, as well as missing values (n) as indicated. Five cases received thrombolysis: three of these were in the subtype of cryptogenic stroke, one was in the LAA group, and one in the cardioembolic stroke group. Oral anticoagulation (OAC) was warfarin. NA, not applicable. ns, not significant. ^a^ Note that the control values were only taken once, i.e., at the inclusion. LDL, low-density lipoprotein; BMI, body mass index; hsCRP, high-sensitivity C-reactive protein; HOMA-IR, Homeostasis model assessment of Insulin Resistance; LAA, large artery atherosclerosis; SAO, small artery occlusion. * *p* < 0.05, ** *p* < 0.01, *** *p* < 0.001.

**Table 2 ijms-24-09120-t002:** Crude correlations between baseline parameters and acute or 3-month s-IGFBP-1 in the total stroke population.

Parameter	Correlation vs. Acute IGFBP-1		Correlation vs. 3-Month IGFBP-1	
	(r)	(*p*)	(r)	(*p*)
Age	0.17	**<0.001**	0.19	**<0.001**
HOMA-IR	−0.33	**<0.001**	−0.26	**<0.001**
BMI	−0.25	**<0.001**	−0.28	**<0.001**
NIHSS	−0.18	**<0.001**	−0.04	0.46
Current smoking	0.11	**0.016**	0.10	**0.035**
hsCRP	0.034	0.49	0.022	0.65
Hypertension	0.001	0.99	−0.50	0.30
LDL	−0.10	**0.032**	0.034	0.50
Diabetes	0.13	**0.006**	0.13	**0.007**

Crude correlations, according to the method of Spearman (rho values, r; and *p*-values, *p*), are shown for each parameter vs. N/A, not applicable; NIHSS, National Institutes of Health Stroke Scale; NS, not significant; LDL, low-density lipoprotein; HOMA-IR, Homeostatic model assessment of IR; BMI, body mass index; hsCRP, high-sensitivity C-reactive protein. *p*-values < 0.05 are indicated in bold.

**Table 3 ijms-24-09120-t003:** Odds ratios (ORs) and 95% confidence intervals (CIs) for the risk of poor functional outcome (mRS ≥ 3) after 3 months, 2 years, and 7 years per log increase of acute IGFBP-1 and per log increase of IGFBP-1 after 3 months in the total stroke population.

Total Stroke Population	Acute IGFBP-1			3 Months IGFBP-1		
2 years mRS		All/mRS ≥ 3 (n)			All/mRS ≥ 3 (n)	
Crude	1.10 (0.62–1.97)	447/91	0.74	3.11 (1.57–6.17) **	428/78	0.001
Multivariate model A	0.93 (0.51–1.70)		0.81	2.76 (1.37–5.59) **		0.005
Multivariate model B	0.75 (0.39–1.45)		0.39	2.38 (1.10–5.15) *		0.028
Multivariate model C	1.59 (0.74–3.40)		0.23	3.41 (1.37–8.52) **		0.009
7 years mRS (including mRS 6)						
Crude	2.17 (1.24–3.79) **	360/128	0.007	5.22 (2.69–10.1) ***	347/117	<0.001
Multivariate model A	1.97 (1.11–3.48) *		0.02	4.73 (2.43–9.22) ***		<0.001
Multivariate model B	1.41 (0.75–2.63)		0.28	3.68 (1.79–7.59) ***		<0.001
Multivariate model C	2.88 (1.40–5.92) **		0.004	5.69 (2.53–12.8) ***		<0.001
7 years mRS (excluding mRS 6 = dead)						
Crude	1.67 (0.82–3.41)	300/68	0.16	4.24 (1.91–9.39) ***	297/67	<0.001
Multivariate model A	1.42 (0.67–2.97)		0.36	3.86 (1.71–8.69) **		0.001
Multivariate model B	1.34 (0.61–2.98)		0.47	3.63 (1.49–8.81) **		0.004
Multivariate model C	3.57 (1.36–9.40) *		0.01	5.12 (1.82–14.4) **		0.002

ORs and 95% CIs were calculated using binary logistic regression. Data are shown as OR (95% CI), respective number of patients included in the analysis (n of the total group of ischemic stroke patients and n of patients with poor outcome, i.e., mRS ≥ 3) and *p*-values. Model A: adjustment for age and sex. Model B: age, sex, and cardiovascular risk factors (BMI, hypertension, LDL, smoking, and diabetes). Model C: age, sex, cardiovascular risk factors, and stroke severity. * *p* < 0.05, ** *p* < 0.01, *** *p* < 0.001.

**Table 4 ijms-24-09120-t004:** Hazards ratios (HRs) and 95% confidence intervals (CIs) for the mortality risk per log increase of acute IGFBP-1 and per log increase of IGFBP-1 after 3 months in the total stroke population.

Total Stroke Population	Acute IGFBP-1		3 Months IGFBP-1	
	Hazard Ratio	*p*	Hazard Ratio	*p*
Crude	2.09 (1.22–3.56) ***	0.007	3.26 (1.82–5.84) ***	<0.001
Multivariate model A	1.81 (1.04–3.16) *	0.037	3.10 (1.68–5.73) ***	<0.001
Multivariate model B	1.28 (0.72–2.26)	0.404	1.99 (1.07–3.71) *	0.031
Multivariate model C	1.38 (0.78–2.47)	0.27	2.00 (1.07–3.73) *	0.030

HRs were calculated using Cox proportional hazards regression. Data are shown as HR (95% CI) and *p*-values. Model A: adjustment for age and sex. Model B: age, sex, and cardiovascular risk factors (BMI, hypertension, LDL, smoking, and diabetes). Model C: age, sex, cardiovascular risk factors, and stroke severity. BMI, body mass index. * *p* < 0.05, *** *p* < 0.001.

**Table 5 ijms-24-09120-t005:** Hazards ratios (HRs) and 95% confidence intervals (CIs) for the mortality risk (A) per log increase of acute IGFBP-1 and (B) per log increase of IGFBP-1 after 3 months for the main IS etiological subtypes.

(A) Per Log Increase of Acute s-IGFBP-1				
Group of Ischemic Stroke (TOAST)	Crude		Age and sex adjusted	
	Hazard Ratio	*p*	Hazard Ratio	*p*
large vessel disease	5.77 (2.00–16.6) **	0.001	4.86 (1.61–14.7) **	0.005
small vessel disease	1.69 (0.29–9.85)	0.56	1.60 (0.26–9.76)	0.61
cardioembolic stroke	1.91 (0.50–7.31)	0.33	1.14 (0.41–3.18)	0.80
cryptogenic stroke	18.0 (0.61–529)	0.098	1.49 (0.34–6.48)	0.60
**(B) Per Log Increase of 3 Months s-IGFBP-1**				
Group of ischemic stroke (TOAST)	Crude		Age and sex adjusted	
	Hazard Ratio	*p*	Hazard Ratio	*p*
large vessel disease	2.71 (0.77–9.60)	0.12	1.90 (0.49–7.34)	0.35
small vessel disease	1.48 (0.27–8.25)	0.65	1.34 (0.22–8.28)	0.75
cardioembolic stroke	3.80 (1.32–10.9) *	0.013	4.76 (1.42–16.0) *	0.012
cryptogenic stroke	5.29 (1.27–22.0) *	0.022	5.53 (1.12–27.3) *	0.036

HRs were calculated using Cox proportional hazards regression. Data are shown as HR (95% CI) and *p*-values, crude and after adjustment for age and sex. * *p* < 0.05, ** *p* < 0.01.

## Data Availability

The data presented in this study are available on reasonable request from the corresponding author. The data are not publicly available due to legal restrictions regarding privacy and ethical issues.

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
