# Peer review of "Serum IGFBP-1 Concentration as a Predictor of Outcome after Ischemic Stroke—A Prospective Observational Study"

_ijms, 2023, doi:10.3390/ijms24119120_

Round 1
Reviewer 1 Report
In the submitted manuscript by Åberg et al., the authors studied the role of serum IGFBP-1 (s-IGFBP-1) in predicting the clinical outcome of ischemic stroke. Their findings suggest that higher s-IGFBP-1 levels after 3 months post-stroke were associated with an increased risk of poor long-term functional outcome and post-stroke mortality, whereas higher acute s-IGFBP-1 was only associated with poor functional outcome after 7 years. Overall, the data are solid, and the reported findings are meaningful for the field. However, the study could be further strengthened by addressing the following points:
1. The rationale for separating acute and 3-month s-IGFBP-1 levels into two independent factors is somewhat unclear. How the authors chose the time point of 3 months? I would advise explain the logic in the text.
2. The study claimed no significant difference between s-IGFBP-1 levels of healthy controls and stroke patients in the acute phase. However, higher acute s-IGFBP-1 seems to predict poor outcome, especially in deceased patients. To gain more insights, can the authors further investigate subgroups of patients?
Author Response
Response from authors:
- The reasons for this are now stated in Materials and Methods, 4.5 Biochemical analysis, second sentence.
- Although analysis investigations of subgroups of stroke were not the primary objective of the present study, we have now performed analyses of the associations between s-IGFBP-1 and the risk of mortality in subgroups of ischemic stroke according to the TOAST classification. The groups are smaller, not permitting us to adjust for the same number of covariates, so the conclusions must be tentative. The results of these analyses of the associations between s-IGFBP-1 and the risk of mortality in the stroke subgroups are now presented in the new Table 5. In addition, we have commented on these analyses in the last paragraph of 3. Discussion, in the last sentence of 3.1 Strengths and limitations, and in Materials and Methods, 4.6 Statistical evaluation, last paragraph.

Reviewer 2 Report
I read with interest the manuscript written by Aberget al., regarding the predictive role of Serum IGFB-1 in the outcome of ischemic stroke on a very large population on long-term follow up. The authors results underlying that high acute s-IGFBP-1 was associated with poor functional outcome after 7 years and s-IGFBP-1 after 3 months was an independent predictor of poor long-term functional outcome and post-stroke mortality.
Overall, the manuscript is well written, well structured, making it easy to read, and the main idea of the article is of the interest. I want to congratulate the authors for their work.
Author Response
Response from authors:
Many thanks for this positive evaluation of the manuscript.

Reviewer 3 Report
The present manuscript propose that the serum IGFBP-1 might be a good predictor of outcome after isquemich stroke. The study have a sufficient patients included to detect statistical significant differences to conclude that the IGFBP-1 might be a good biomarker. The introduction and discussion is well write and with sufficient information to introduce the issue and to have an overview of all studies published about it. However, I have some questions about the results and its representation in the manuscript.
First of all, in table 1, there are the "n" of each group, but, is there any missing value in some parameter analysed?
Secondly, the authors comment that logarithmic transformation was performed to normalized the values of IGFBP-1. However, is not totally clear if the transformation was done before obtain figure 1 or not. To obtain the figure 1, which values of IGFBP-1 were used? I suggest to use the logarithmic values to normalize the data. Have the levels of acute IGFBP-1 a normal distribution, before and after logarithmic transformation? Depending of the distribution the authors might be to use one statistical test or another. In addition, how many values are in each timepoint? I suggest to add the "n" of each timepoint in the graph or dots in the graph to visualizehow many values are in each group.
Then, I suggest to add some figure or table to make more clear the results explained in the 2.3 section about "Bivariate correlations". In this section there are a lot of p-values and coefficients of regression and the table or figure will help to underly the conclusions of all analysis. If the authors do not considered necessary to add this information as a main figure or table I suggest to add this information in a supplementary table and/or cite table 1 in the 2.3 section.
Finally, which are the exclusion criteria? All 129-128 samples exclude were due to inadequate blood samples? The authors made some quality control?
Author Response
Response from authors:
Information regarding missing values in patients and controls has been added in Table 1.
There are few missing values in most of the acute parameters that were analyzed. However, there were more missing values in of terms of LDL and HOMA-IR. Regarding LDL, which had the highest amount of missing data and was included in the regression models, we carried out a data imputation, which is described in Materials and Methods, 4.6 Statistical evaluation, last paragraph. We did not perform any data imputation in terms of HOMA-IR as this variable was not included in the regression analyses.
We have added the number “n” of each timepoint in Figure 1.
The levels of acute IGFBP-1 were not normally distributed before the logarithmic transformation (Materials and Methods, 4.6 Statistical evaluation, first paragraph, second sentence). However, after logarithmic transformation, s-IGFBP-1 was normally distributed (Materials and Methods, 4.6 Statistical evaluation, first paragraph, last sentence). For the knowledge of the Reviewer, in terms of Figure 1, we have performed analyses using logarithmically transformed acute s-IGFBP-1, and also in these analyses, the levels of acute s-IGFBP-1 were not related to when the blood sample was taken.
In Figure 1, the values of s-IGFBP-1 were not logarithmically transformed, but we believe that showing medians, 25th–75th percentiles, and ranges is a valid way of presenting s-IGFBP-1. However, in Table 3-5 (the Tables have a new nomenclature due to a new Table 2 and a new Table 5), logarithmic transformation of s-IGFBP-1 levels were performed before the regression analyses were conducted.
This is a valuable suggestion from the Reviewer. Therefore, we have added a new table in the manuscript. In this table (new Table 2), the results of the bivariate correlation analyses are given. In addition, we have rewritten this section in Results (2.3. Bivariate correlations) in order to facilitate for the reader.
The exclusions were due to inadequate blood samples. There were no other inclusion/exclusion criteria than those described in Materials and Methods, 4.1 Patients and controls. Regarding the overall quality control, the measurements of s-IGFBP-1 were performed at the same time point (Materials and Methods, 4.5 Biochemical analysis). Furthermore, both the inter-assay CV and the intra-assay CV are stated in Materials and Methods, 4.5 Biochemical analysis. Finally, due to the relatively high inter-assay variation for s-IGFBP-1, a correction factor was used, obtained by having three original serum samples in each 96-well plate (Materials and Methods, 4.5 Biochemical analysis).

Reviewer 4 Report
The problem of stroke occurrence in patients with or without previously diagnosed cardiac disease is a significant clinical problem. A patient who has suffered a stroke requires chronic use of pharmacological secondary prevention. For this reason, the search for biochemical markers that are predictors of outcome is very important. Study designed correctly. Vetches presented clearly, well discussed.
In order to increase the quality of work, please:
1st extension of table 1:
- adding information about past smokers
- information about the diagnosis of atrial fibrillation
- information on previous occurrence of atrial fibrillation
- if AF, what type of anticoagulation was used (VKA/DOAC)
- whether patients were taking antiplatelet drugs
2. to be discussed
- how effective was the anticoagulation? was the INR within the therapeutic range? Were DOACs applied correctly (adherence)?
- whether Xa or dabigatran dTT activity was evaluated for DOAC (an 8-year observational study on therapeutic monitoring of direct oral anticoagulants has already been published)
- what were the coexisting diseases (except diabetes listed in the table)
- was fibrinolysis or thrombectomy performed?
- whether it was necessary to administer idarucizumab or andexanat-alpha before the above-mentioned treatment
Author Response
Response from authors:
Unfortunately, regarding smoking, we only have information about current smoking or no smoking/smoking cessation > 1 year ago. This is now stated in Materials and Methods, 4.4 Assessment of covariates and ethical considerations, first paragraph, last sentence. Although this was not the focus of the present study of s-IGFBP-1, we agree that it is of value to add information regarding cardiac diseases and medication. Data regarding atrial fibrillation at inclusion are now presented in Results, 2.1 Baseline characteristics and s-IGFBP-1 in patients and controls, second sentence as well as in Materials and Methods, 4.4 Assessment of covariates and ethical considerations, first sentence. Also, information in terms of atrial fibrillation and oral anticoagulant (OAC) therapy has been added in Table 1.
As stated in Materials and Methods, 4.1 Patients and controls, third sentence, SAHLSIS was commenced in the late 1990s. Thus, the inclusion of the patients took place 20 years ago, and the treatment regimens administered to stroke patients have changed since then. We have data over OAC, i.e., warfarin was mainly used (Table 1 and footnote of Table 1). However, we do not have reliable information about antiplatelet drugs (mainly ASA when the patients were included). Moreover, thrombolysis was rarely administered and even fewer thrombectomies took place. This is now stated as a study limitation in Discussion, 3.1 Strengths and limitations, second and third last sentences.
2. We agree that this is of interest. Although the objectives of our study of s-IGFBP-1 were not focused on these aspects, we have been able to include some additional information. In Table 1, hypertension is listed, and we have also added data in terms of atrial fibrillation in this table. We do not have data in terms of INR and adherence to treatment of oral anticoagulants prior to the stroke event in our ischemic stroke patients. SAHLSIS was commenced in the late 1990s, and thus, the inclusion of the patients took place 20 years ago. Generally, studies of Swedish patients in that era, present a high proportion of INR in the therapeutic range, as exhibited in for instance Lancet 2003 Olsson et al (PMID 14643116). When our study was initiated, fibrinolysis was not a common procedure we have added data over the five patients that received fibrinolysis in Table 1 (in the footnote). Furthermore, this is stated as a study limitation in Discussion, 3.1 Strengths and limitations, latter part.
DOACs were not used in the present study as the first DOAC was registered in Sweden in 2011, i.e., when the inclusion of patients in our study had been completed. Furthermore, as a consequence, Xa or dabigatran dTT activity was not evaluated in the present study. Finally, it was not necessary to administer idarucizumab or andexanat-alpha as our study was initiated more than 20 years ago (before the DOAC era). In Discussion, 3.1 Strengths and limitations, there is now a brief discussion of the changes in treatment regimens that have occurred after the present study was initiated.

Round 2
Reviewer 4 Report
The authors of the manuscript have extended some of the wording, which makes it considered for publication.